# A mid-infrared lab-on-a-chip for dynamic reaction monitoring

Borislav Hinkov [1] ✉, Florian Pilat[1], Laurin Lux[2], Patricia L. Souza [1,3], Mauro David [1], Andreas Schwaighofer [2], Daniela Ristanić[1], Benedikt Schwarz [1], Hermann Detz [1,4], Aaron M. Andrews [1], Bernhard Lendl [2] & Gottfried Strasser [1]

Mid-infrared spectroscopy is a sensitive and selective technique for probing molecules in the gas or liquid phase. Investigating chemical reactions in bio-medical applications such as drug production is recently gaining particular interest. However, monitoring dynamic processes in liquids is commonly limited to bulky systems and thus requires time-consuming offline analytics. In this work, we show a next-generation, fully-integrated and robust chip-scale sensor for online measurements of molecule dynamics in a liquid solution. Our fingertip-sized device utilizes quantum cascade technology, combining the emitter, sensing section and detector on a single chip. This enables real-time measurements probing only microliter amounts of analyte in an in situ configuration. We demonstrate time-resolved device operation by analyzing temperature-induced conformational changes of the model protein bovine serum albumin in heavy water. Quantitative measurements reveal excellent performance characteristics in terms of sensor linearity, wide coverage of concentrations, extending from 0.075 mg ml⁻¹ to 92 mg ml⁻¹ and a 55-times higher absorbance than state-of-the-art bulky and offline reference systems.

Sensors have entered our daily life on countless levels, from medical diagnostics[1–3], environmental sensing and climate research[4,5] to spectral imaging[6] and security applications[7]. They detect, analyze, and react to all sorts of relevant substances, e.g., potentially hazardous chemicals[8]. While mid-infrared (mid-IR) gas-phase spectroscopy is nowadays well exploited for sensing applications based on quantum cascade (QC) technology[9–11], liquid detection techniques are still in their infancy[12–14]. They include, e.g., trying to address the very broad absorption bands (>10–50 cm⁻¹) in the much higher density medium of liquids[15–17]. This becomes an even more challenging task, when detecting target analytes at (i) very low (ppb- to ppt-) concentration-levels or (ii) rapidly changing concentrations, while investigating chemical reactions or conformational changes of molecules. Desirable characteristics for sensors monitoring dynamic processes in the liquid phase include rapid response times, high sensitivity and specificity, as

well as the ability to analyze wide dynamic concentration-ranges in microliter sample sizes.

Consequently, it is highly beneficial for a high sensor specificity to target the spectral fingerprint region of fundamental molecule absorptions in the mid-infrared spectral range (~500–1700 cm⁻¹[18,19]), and in particular the region of the protein amide I band (~1600–1700 cm⁻¹) in the case of protein analysis[20].

The sensitivity of a sensor depends on its noise performance and slope of calibration line. In spectroscopic techniques based on the Beer-Lambert law, the sensitivity can be tailored by maximizing the effective interaction length of the light within the sample. However, typical mid-IR absorption length values in aqueous solution lie on the low micrometer-scale for the existing techniques and often use bulky devices[9,14,21]. Consequently, high-power light sources such as QC lasers (QCLs) and high-performance detectors, like QC detectors (QCDs), are

[1]Institute of Solid State Electronics & Center for Micro- and Nanostructures, TU Wien, 1040 Vienna, Austria. [2]Institute of Chemical Technologies and Analytics, TU Wien, 1060 Vienna, Austria. [3]LabSem-CETUC, Pontifícia Universidade Católica do Rio de Janeiro, Rio de Janeiro, Brazil. [4]CEITEC, Brno University of Technology, Brno, Czech Republic. ✉e-mail: borislav.hinkov@tuwien.ac.at

favorable tools for improvements. They allow addressing real-world applications in mid-IR liquid-phase spectroscopy, and are able to probe sample film thicknesses far beyond a few micrometers, thus enabling simplified and more robust sample handling[8,13,22].

In contrast to sensor specificity and sensitivity that were already addressed by first experiments in literature[23], we want to demonstrate a concept that shows significant progress on two additional critical features:

(i) Dynamic processes, such as those found in chemical reactions[24] or conformational changes, i.e., structural changes of a molecules three-dimensional structure[13], reveal important characteristics that have to be analyzed with high temporal resolution for their adequate investigation. An in situ sensor for label-free real-time measurements is the ideal tool for monitoring those analyte-changes, completely avoiding time-consuming offline analytics.

(ii) The ability to on-chip analyze minute amounts of liquids enables detection schemes for real-world applications through sensor miniaturization. This includes online measurements of microliter-samples, only minimally interfering with chemical processes.

In this work, we present a fully monolithic integrated mid-IR sensor, that combines all of the above features into a single, miniaturized device. Through the combination of the laser, interaction region, and detector on one chip, and avoiding typical diffraction limitations of conventional chip-scale photonic systems[6,25] by exploiting plasmonic waveguides[26–28], we realize a fingertip sized ($<5 \times 5$ mm²) next-generation rapid liquid sensor. Simulation results confirm the preservation of plasmonic capabilities in a liquid environment and enable the use of spectrally optimized QCLDs, i.e., devices that emit and detect similar-wavelength photons[29]. We perform two types of measurements in our study. We determine the sensor calibration line and perform thermal denaturation experiments, monitoring the related change of the protein secondary structure, both of bovine serum albumin (BSA)[12,13,30–33] in a $D_2O$ matrix. Since our work includes an analysis of the sensor performance using optical Finite Element (FEM) simulations with the commercial software COMSOL, we can also theoretically confirm its excellent suitability for in situ operation in a liquid matrix. We then determine experimentally important analytical figures-of-merit including: (1) LOD, (2) sensor-linearity with analyte concentration, (3) the accessible concentration-range and sample volume of our sensor, and (4) robustness against direct exposure to the analyte. We complete our study by demonstrating the operation of our QCLD sensor when immersed into normal water ($H_2O$), as biophysically native and most relevant matrix. As a direct consequence of the fully absorbed intensity of the optical mode in water, due to the large effective penetration depth of the sensor, we can simultaneously use the remaining measured detector signal from this experiment for electrical crosstalk correction.

## Results

### Measuring bovine serum albumin

We aim to accurately characterize our QCLD sensor and extract its relevant figures-of-merit, together with demonstrating its capability to monitor changes in the secondary structure of proteins in real time. For these experiments, we use BSA as a water-soluble monomeric protein (see Fig. 1a and Supplementary A). BSA is frequently used in fundamental biophysical studies, such as the investigation of thermal denaturation[30,34,35] that leads to changes in mid-IR absorption in the amide I region of proteins. Investigations of thermal denaturation of BSA were previously regularly performed in both, $H_2O$ and $D_2O$[30–32,34,35], providing a rich data set, especially for those experiments in heavy water.

Heavy water is often employed for protein analysis to avoid overlap of the HOH bending band of water with the protein amide I region in mid-IR spectroscopy. This creates an open spectral window in this range with reduced absorption, enabling larger interaction lengths[12,30–32,35–42]. Generally, heavy water can provide some deviations from fully native biophysical conditions. It was found that exposure of proteins to $D_2O$ may influence the length and strength of hydrogen bonds and may cause changes in protein dynamics[43] as well as protein denaturation[44]. But such different properties of $D_2O$ can also be exploited in the analysis of specific biological samples, e.g., for significantly slowing down protein-solvent interactions in living cells for in vitro experiments[45].

For the here reported experiment in the thermal denaturation of BSA, it was found that the structural transitions of BSA were similar in $H_2O$ and $D_2O$, with the single difference of a lower transition temperature in heavy water[32]. Working in $D_2O$ thus allows us to record data with a high signal-to-noise ratio and to exploit the full potential of our sensor concept in dynamic reaction monitoring. It also enables to much better harness the plasmonic-sensing concept, which allows optical mode propagation on the tens-to-hundreds of micrometers length scales. Our on-chip sensor features a sample interaction length of ~48 μm, which facilitates the analysis of BSA in $D_2O$ over a wide concentration range, covering more than three orders of magnitude, from ~75 μg ml⁻¹ to >92 mg ml⁻¹. In contrast, using highly absorbing $H_2O$ buffer typically limits path lengths to a maximum of 10 μm in the case of low-intensity FTIR-based experiments[46] and to ~25 μm[47] when performing high-intensity QCL-based transmission measurements,

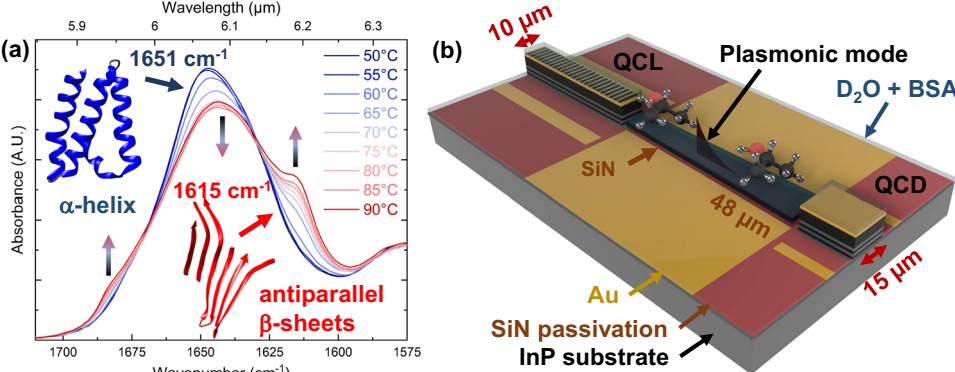

**Fig. 1 | FTIR spectrum of bovine serum albumin (BSA) & schematic of the QCLD device. a** Attenuated total reflection Fourier-transform infrared spectrometer (ATR-FTIR) reference measurement of the thermal denaturation process of BSA, analyzed in the range of the amide I′ band between 50 °C (blue) and 90 °C (red). The temperature-induced transition from $\alpha$-helix (1651 cm⁻¹, blue) to $\beta$-sheet (1615 cm⁻¹, red) is indicated. **b** On-chip sensor concept including indicated plasmonic mode.

Emitter (QCL, 10 μm wide) and detector (QCD, 15 μm wide) are connected through a 48 μm long tapered SiN-based plasmonic waveguide. The whole sensor is submerged into the sample solution ($D_2O$ + BSA), which is shown by the blue transparent layer on the chip. The gold layer (plasmonic waveguide and electrical contacts) is indicated in gold color, the SiN passivation and dielectric loading layer are shown in brown and the InP substrate is indicated in dark gray.

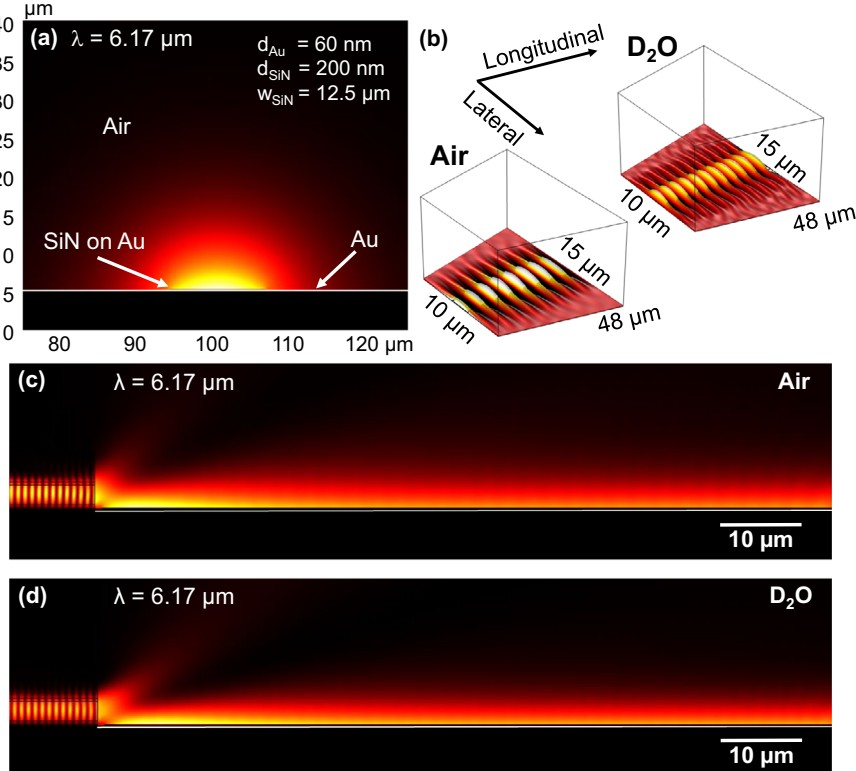

**Fig. 2 | FEM-based simulations at 1620 cm⁻¹ (=6.17 μm). a** The mode cross-section of the SiN on Au DLSPP waveguide ($n_{SiN}$ = 1.79, dimensions in inset: $d_{Au}$: gold thickness, $d_{SiN}$: SiN thickness and $w_{SiN}$: width SiN slab), **b** 2D topview simulation along the tapered 48 μm DLSPP waveguide between QCL and QCD in air (left) and $D_2O$ (right) as well as longitudinal cross-section profile along the 48 μm DLSPP waveguide conducted in: **c** air and **d** $D_2O$. The white line in **a**, **c**, and **d** represents the Au plasmonic layer.

with the consequence of a significantly reduced limit of detection (LOD).

## Optical mid-IR lab-on-a-chip

Quantum cascade technology hosts very powerful and versatile tools for mid-IR gas- and liquid-phase spectroscopy[2,14,26,48–50]. The ability to operate an unbiased QCL as a high-performance photodetector[51–53], enabled the realization of monolithic integrated QCL and QCD, noted as a QCLD device[29]. It features excellent spectral overlap between laser and detector[29]. In this work, we unlock the full potential of the QCLD concept for optical lab-on-chip applications, suitable in the analysis of proteins in the spectral range of the amide I′ band[13,14,32,35].

The QCLD used in this work is based on a bound-to-continuum active region (AR) design, optimized for the same emission and detection wavelength. To target the amide I′ spectral band, hosting the absorption range of BSA in $D_2O$[32,35], it is designed to operate around 6.5 μm wavelength. In particular, the AR is built from $In_{0.53}Ga_{0.47}As$/$In_{0.52}Al_{0.48}As$ quantum wells/barriers in a total of 37 cascades, which are grown lattice-matched to the n-InP substrate by molecular beam epitaxy (MBE) and sandwiched in a waveguide structure. For selecting individual spectral emission modes that target narrow wavelength ranges within the broad absorption features of the analyte of ~20–40 cm⁻¹, a distributed feedback (DFB) grating[54–56] is implemented into the upper cladding of the QCL waveguide structure[57,58] (details on AR[26]: and DFB grating: Supplementary B). As shown by Ristanić et al.[27], this leads to linewidths on the ~MHz-scale and below[56] and improves noise and emission fluctuations in pulsed lasers[59,60].

We use standard Fabry-Pérot (FP) ridge waveguides for laser (~2.5 mm long) and detector (~200 μm long), separated by a ~48 μm long dielectric-loaded surface plasmon polariton (DLSPP) waveguide (200-nm-thick slab of SiN on top of a 60-nm thick Au bottom layer, see

Fig. 1b). The latter is tapered from 10 μm wide at the QCL to 15 μm at the QCD. Due to the low electrical conductivity of the analyte, we can directly submerge our sensor into the liquid without additional protective coatings. We further increased the QCLD-sensor sensitivity by tackling the dominant technical device noise sources of temperature fluctuations[61,62] and electrical crosstalk known to compact on-chip geometries[17,63]. The former was addressed through on-chip temperature measurements and the latter by electrical contacts with increased separation and a post-experiment crosstalk correction. The detailed spectral emission and detection characteristics of the QCLD are shown in Supplementary Fig. 1.

## Dielectric-loaded plasmonic waveguides

The spectral performance of a plasmonic waveguide is dominated by its structural and material characteristics (complex refractive index $n$) at the target wavelength $\lambda$ including the surrounding host medium. It was shown that for thin DLSPP waveguides, a remarkable portion of >96% of the mode is guided outside the waveguide (DLSPP thickness ≪ wavelength), penetrating its surrounding dielectric medium, like, e.g., air[26]. This makes such waveguides highly suitable for liquid spectroscopy, as their propagation properties are susceptible to their surrounding medium.

For analyzing our SiN-based DLSPP waveguide when exposed to $D_2O$ and BSA in $D_2O$, we simulate the propagation of the plasmonic mode using the eigenmode solver of the FEM-based commercial software COMSOL (v.5.5). We focus on the two wavelengths of interest: 6.26 μm (1597 cm⁻¹, concentration series) and 6.17 μm (1620 cm⁻¹, BSA denaturation experiment). Figure 2a shows the transverse mode profile in air at 6.17 μm with $n_{SiN}$ = 1.79.

The refractive index $n_{SiN}$ is obtained from mid-IR ellipsometer measurements shown in Supplementary Fig. 2, indicating similar

**Table 1 | Figures-of-merit of the DLSPP waveguide in air**

| Figure-of-merit | | |
|---|---|---|
| Wavelength (cm$^{-1}$) | 1597 | 1620 |
| Propagation Length $L_p$ (µm) | 1814 | 1692 |
| $n_{eff}$ | 1.002 | 1.002 |
| Losses (dB mm$^{-1}$) | 2.3937 | 2.5672 |
| Losses (dB per 48 µm) | 0.1149 | 0.1232 |

The obtained values are calculated with the FEM-based commercial software COMSOL (v5.5) at both wavelengths of interest.

results to literature[64]. The obtained values at both wavelengths, i.e., 1597 cm$^{-1}$ and 1620 cm$^{-1}$, for the propagation length $L_p$ (1/e-decay distance in µm), effective mode index $n_{eff}$ and losses (in dB mm$^{-1}$ and dB per 48 µm = plasmonic section between QCL and QCD) are shown in Table 1. The propagation length is 7% lower at shorter wavelength, a consequence of the slightly more suitable plasmonic waveguide geometry for longer wavelengths[28]. Still, $L_p$ is ≥1.7 mm, which corresponds to losses below 0.13 dB for a 48 µm waveguide section, confirming the low-loss characteristics of our DLSPP waveguides in air. Mode profile and $n_{eff}$, show only negligible differences at the two wavelengths.

In the following, we compare the 6.17 µm longitudinal mode profiles in air to the case of D$_2$O as surrounding medium. As Fig. 2 shows, the mode is very well confined in air (see Fig. 2b, c), and we observe similar laser-waveguide coupling in D$_2$O (see Fig. 2d, ref. 40). This is a remarkable result, because the refractive index of air ($n_{air} \approx 1$) is significantly lower than that for D$_2$O of $n_{D_2O} = 1.3$[40]. Still, the mode remains very well confined and guided from laser to detector, showing the excellent suitability of the DLSPP waveguide for liquid spectroscopy. Adding the BSA to the D$_2$O has only a negligible effect on the refractive index (e.g., $\Delta n \sim 10^{-4}$ for 0.25–2% m v$^{-1}$[40]) as compared to pure deuterium oxide.

Finally, we investigated the influence of the D$_2$O on the plasmonic mode by topview 2D-simulations of the tapered DLSPP waveguide, displaying an horizontal cut at 60 nm above the SiN layer. As shown in Fig. 2b there are only minor differences when comparing air (left) to D$_2$O (right), thus, also only minimal reduction of the lateral mode confinement.

### On-chip concentration series in submersion configuration

One common way to measure the absorbance of liquids in the mid-IR spectral range is based on attenuated total reflection (ATR) spectroscopy[65]. In this technique, the sample is placed on the surface of an optically dense ATR element. Impinging infrared light under a certain angle is reflected at the interface towards the sample, minimally penetrating it with its evanescent field.

In such a configuration, the absorbance A of a liquid with the effective layer thickness $d_{eff}$ can be obtained using the Beer-Lambert law[9,66]: $A = d_{eff} \cdot e \cdot c$, with the molar decadic absorption coefficient $e$ and the concentration of the analyte $c$. An experimental A vs c curve therefore shows a linear dependence[65,66] with the slope given by the product of $d_{eff} \cdot e$.

The effective path length of our QCLD sensor was determined by comparing its absorbance with that of the reference single-reflection attenuated total reflection Fourier-transform infrared (ATR-FTIR) interferometer measurements. For the ATR accessory we obtain: $d_{eff} = 0.838$ µm at 1597 cm$^{-1}$, determined in H$_2$O ($e = 10.9$ L mol$^{-1}$ cm$^{-1}$)[67].

We calibrated the performance of our QCLD-sensor through a concentration series of BSA in D$_2$O, obtaining its A vs c curve. We compare it to results from other current state-of-the-art measurement techniques, including circular dichroism (CD) and FTIR spectroscopy[13], as well as ATR-based sensors[9], including, e.g., fiber-based ATR-FTIR spectroscopy[12]. In addition, we extract important performance values of the sensor such as the LOD and the effective path length within the probed analyte $d_{eff}$. This is in contrast to previous studies, mainly performing a qualitative analysis[34,35,40,68].

Figure 3 shows the concentration measurement setup. It includes a ~50 ml stock solution of BSA in D$_2$O at a fixed concentration of 150 mg ml$^{-1}$. Using a peristaltic pump (Ismatec Reglo ICC, 3 channels, 8 rolls) we continuously add 1 ml of the stock solution to a second beaker filled with 30 ml of pure D$_2$O (99.9 atom % D). The QCLD-sensor is directly submerged into the D$_2$O beaker showing its robustness towards direct exposure to the protein-containing solution. For this experiment we bias QCL 1 emitting at 1597 cm$^{-1}$, measure the signal of its corresponding QCD 1 and analyze it with a 350 MHz oscilloscope (Teledyne LeCroy HDO4034 2.5 GSPS). In parallel, we monitor the temperature of the liquid with two previously characterized temperature probes: (i) we bias neighboring QCL 2 at 0.5 mA and monitor its temperature-induced resistance change using a sourcemeter (Keithley 2400 series), as a fast on-chip temperature-probe and (ii) we additionally submerge a temperature probe (Pt100) into the liquid.

Figure 4 shows the resulting A vs c curves at 1597 cm$^{-1}$ and room temperature measured in absorbance units (AU) with our QCLD-sensor (blue squares) and the ATR-FTIR reference sensor (violet circles). The

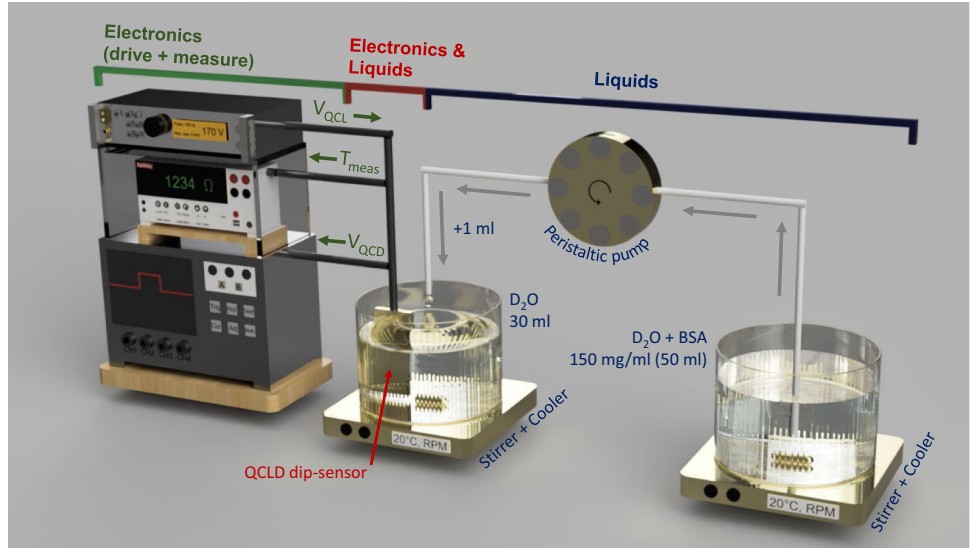

**Fig. 3 | Concentration measurement setup.** A peristaltic pump continuously pumps the stock solution (50 ml BSA in D$_2$O at 150 mg ml$^{-1}$) to the measurement beaker, initally filled with pure D$_2$O. The QCLD-sensor is directly submerged in this second beaker to monitor the concentration changes.

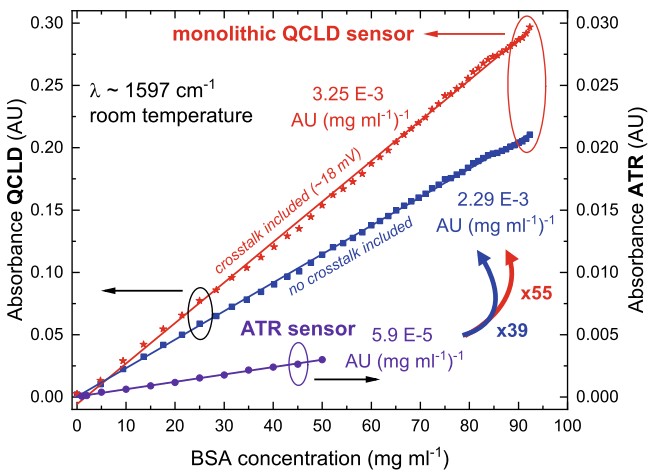

**Fig. 4 | Absorbance vs concentration measurements at 1597 cm⁻¹.** Results in absorbance units (AU) of the QCLD sensor for BSA in D₂O with (red stars) and without (blue squares) 18-mV crosstalk correction (left scale) and in comparison to the single-reflection ATR-FTIR system (violet circles, right scale). The right scale is divided by a factor of 10 as compared to the left one for better visibility of the ATR-FTIR signal.

absorbance is obtained by normalizing the BSA signal to its previously measured D₂O-only baseline and taking the decadic logarithm of this value. As expected for a sensor following the Beer-Lambert law, we obtain a linear calibration line[66]. We want to stress our ability to probe 48 μm of solution for a wide range of BSA-concentrations from <100 μg ml⁻¹ to >92 mg ml⁻¹. In contrast, such experiments were so far typically performed with large and bulky ATR-FTIR-based systems[31,34].

The ATR-spectra from Fig. 1a reveal that the absorbance $A = 0.00118$ AU (absorbance units) of BSA is pretty low at 1597 cm⁻¹ in a 20 mg ml⁻¹ BSA solution, which enables probing higher concentrations. In contrast, it is preferable to measure at high absorbance, for analyzing the low-concentration range. As shown in Fig. 4 for 85 mg ml⁻¹ and above, we can observe deviations from the linear curve. They result from a low QCD signal for increasing BSA concentrations. We can conclude, that the maximum concentrations we can measure with the sensor is expected in the range of 92 mg ml⁻¹ at 1597 cm⁻¹.

A direct comparison to the ATR-measurement shows, that our on-chip sensor yields a 39 times higher absorbance, clearly outperforming the state-of-the-art ATR-FTIR system. In addition, the ability to submerge our sensor directly into the liquid without additional protective measures, is a clear benefit of our in situ approach, and enables real-time inline-monitoring of chemical reactions in advanced chemical systems. This is in strong contrast to typical state-of-the-art analytical techniques including ATR-FTIR spectroscopy, which are either limited to offline measurements, or to online measurements, when e.g., merging a fluidic cell with the ATR crystal[69] into still more complex and bulky systems.

### Electrical crosstalk & limit of detection

While the compactness of the QCLD-based approach leads to the previously discussed clear advantages of the sensor, we do observe electrical crosstalk as a drawback of the small-footprint integration (typically: <25 mm²). This parasitic effect can be quantified by performing measurements in fully-absorbing deionized (DI) H₂O, and correcting the BSA-measurements for the obtained electrical crosstalk. The result is depicted in Fig. 4 as red stars and yields even 55 times larger absorbance values than the ATR-FTIR setup. This can be used to estimate the effective penetration depth $d_{eff,QCLD}$ through: $55 \times d_{eff,ATR} = 46.09\ \mu m = d_{eff,QCLD}$ (with $d_{eff,ATR} = 0.838\ \mu m$) and agrees very well with the actual length of 48 μm when considering that 96% of the mode is guided outside waveguide.

**Table 2 | Experimental figures-of-merit for ATR-FTIR & QCLD-sensor**

| Sensor/value | λ (cm⁻¹) | $d_{eff}$ (μm) | c ($\frac{mg}{ml}$) | A (AU) | e ($\frac{ml}{mg\cdot\mu m}$) |
|---|---|---|---|---|---|
| ATR-FTIR | ~1597 | 0.838 (meas.) | 20 | ~0.00118 | 7.04 E-5 |
| QCLD | ~1597 | 43.1 (meas.) | 21.4 | ~0.065 | 7.04 E-5 |

The values include: wavelength λ, effective optical penetration-depth $d_{eff}$, analyte concentration c, absorbance A, and absorption coefficient e at 1597 cm⁻¹.

**Table 3 | LOD of the ATR-FTIR reference setup & monolithic QCLD-sensor**

| Sensor/value | Avg. time (s) | Std(t) (mV) | LOD (mV) | Slope m [0–25 mg/ml] | LOD ($\frac{mg}{ml}$), ppm by weight |
|---|---|---|---|---|---|
| ATR-FTIR | 11 | – | – | | ~9, ~9000 |
| QCLD | 11 | 0.031 ⇒ 0.092 | | −0.4 $\frac{mV}{mg/ml}$ | 0.075, 75 |

LOD calculated for BSA in D₂O.

For further quantitative analysis, we can also calculate this value by using the experimental $A$-vs-$c$ data from Fig. 4 in combination with the already introduced formula for the Beer-Lambert law and the previously obtained $d_{eff,ATR} = 0.838\ \mu m$. The result is given in Table 2. The obtained value of $d_{eff,QCLD} = 43.1\ \mu m$ is again in good agreement with the actual length of the plasmonic waveguide of 48 μm.

As another important figure-of-merit in chemical sensing, we determined the LOD of our QCLD-sensor in comparison to the ATR-FTIR setup as given in Supplementary E. In order to evaluate the LOD of the on-chip sensor for rapid time scales, we calculated the standard deviation std(t) for multiple ~11-s long time intervals measured for a 20 mg ml⁻¹ BSA in D₂O solution and determine the corresponding LOD = 0.092 mV (see Table 3, slope from Supplementary Fig. 3): this corresponds to the minimum detectable concentration change of BSA in mg ml⁻¹ by using the measured calibration line in the low-concentration range 0–25 mg ml⁻¹. It results in a minimum retrievable BSA-concentration of: LOD (mg ml⁻¹) = 75 μg ml⁻¹ ⇒ LOD(ppm) = 75 ppm by weight and a coverage of more than three orders of (BSA-) concentrations, between 75 μg ml⁻¹ and 92 mg ml⁻¹. It is worth noting that our submerged sensor is only temperature stabilized through the temperature stability of the liquid, while the on-chip temperature measurement was only used for monitoring purposes, without using it for stabilization measures. For comparison Table 3 shows the LOD of the ATR setup. While the clear advantage of the FTIR-based ATR technique lies in the recording of a full IR-spectrum (400–4000 cm⁻¹) within the measurement time of 11 s, it is a remarkable result that our on-chip sensor has a 120 times lower LOD. The drawback of addressing one single wavelength with our QCLD-based sensor can be significantly mitigated by straightforward implementation of QC-based array concepts[17,70].

Other approaches for protein sensing reported in literature, especially those based on SEIRAS (Surface Enhanced IR Absorption Spectroscopy) combined with traditional ATR-techniques, result in even better LODs than our QCLD (factor of ~2.1 to -18.1). But they rely on more complex schemes, by, e.g., adding different types of Au nanoparticles[71,72]. This is in contrast to the not yet funtionalized surface of our QCLD-sensor, which is part of future work[73,74].

### Monitoring thermal denaturation of BSA in a miniaturized cell

In Fig. 5 we present the setup for the thermal denaturation experiment of BSA. In this experiment, we use a custom-made 60-μl microliter-scale flow cell. The step-by-step measurement routine is described in the "Methods" section.

For monitoring the thermal denaturation process[31], we employed a QCLD-sensor addressing a wavelength of 1620 cm⁻¹. Figure 6 shows

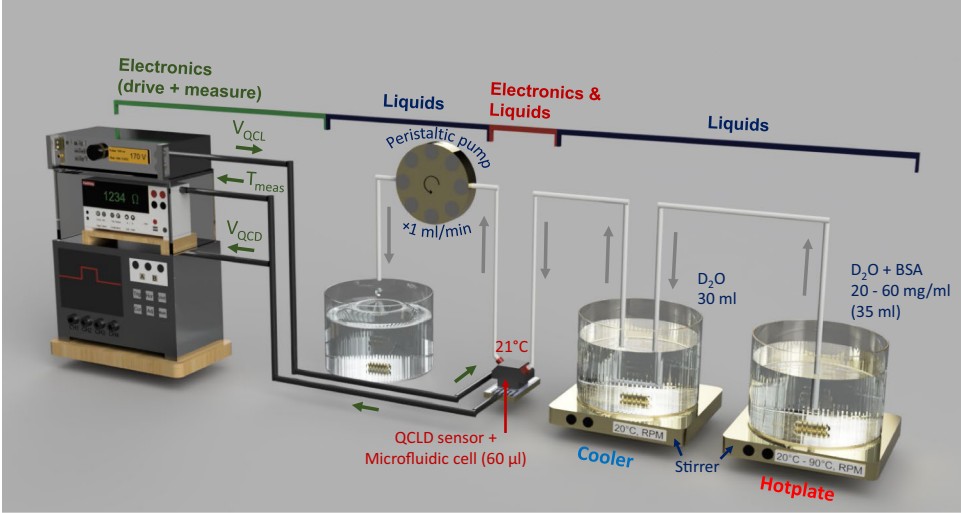

**Fig. 5 | Denaturation measurement setup.** We use a custom-made 60-µl cell for this experiment. The stock solution (35 ml BSA in $D_2O$ at 20, 40, and 60 mg ml$^{-1}$) is constantly heated from room temperature to 90 ℃, while being continuously pumped through a beaker with cooling liquid at 20 ℃ and into the cell containing the QCLD-sensor. After (continuous) measurement it is pumped out and disposed of.

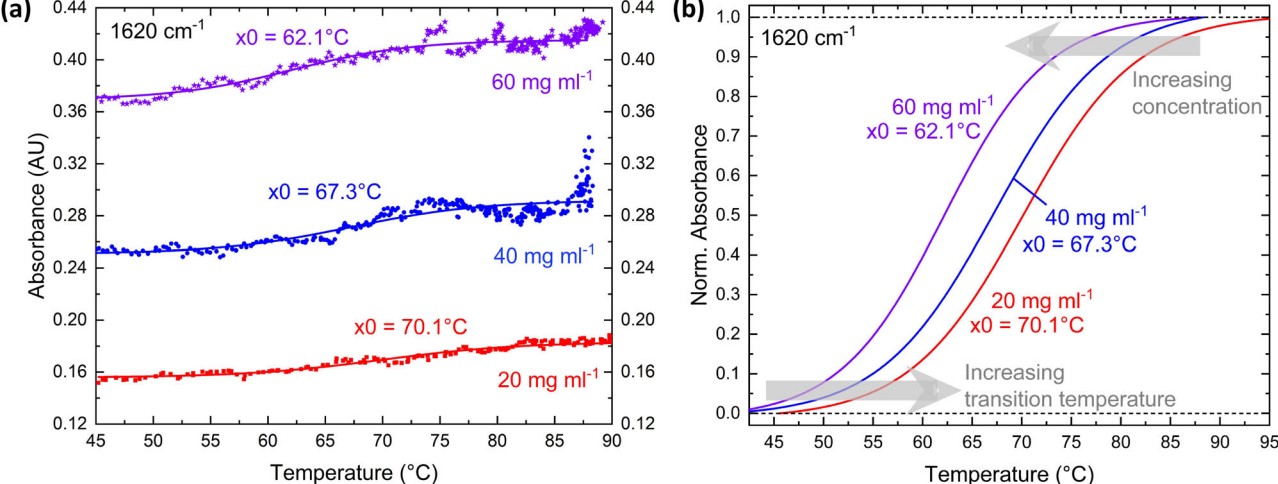

**Fig. 6 | Denaturation measurement results at 1620 cm$^{-1}$.** Investigation of three different concentrations of BSA: 20 (red), 40 (blue), and 60 mg ml$^{-1}$ (violet) and extraction of the transition temperature $x0$. **a** Absolute measurement values (circles) and sigmoidal Boltzmann-Fit curves (solid lines) in absorbance units (AU).

**b** Comparison of the (individually normalized) Boltzmann-fit curves, showing the temperature- and concentration-dependence of the sigmoidal-shaped absorbance curves.

the obtained results. The absolute values in absorbance units (AU) in Fig. 6a include crosstalk correction, as already discussed for the concentration series. To obtain the crosstalk correction value, we use the expected absorbance ratio at the two wavelengths: $A$(1620 cm$^{-1}$) per $A$(1597 cm$^{-1}$). We extract a crosstalk of 6 mV, which is in very good agreement with our sensor analysis at 1620 cm$^{-1}$, revealing that we can measure reasonable denaturation curves to signal levels of about 10 mV.

Figure 6a depicts the absorbance signal for different temperatures between 45 ℃ and ~90 ℃ for the three concentrations 20 mg ml$^{-1}$ (red), 40 mg ml$^{-1}$ (blue), and 60 mg ml$^{-1}$ (violet) at 1620 cm$^{-1}$. We can observe the expected sigmoidal shape from the protein unfolding process[13] for all three investigated concentrations, confirming previous findings from literature[34]. One additional benefit of our online experiment can be found in its very little liquid consumption (pump-rate: ~17 µl s$^{-1}$, microfluidic cell volume: ~60 µl).

For quantitative evaluation of the measurements and comparison to literature, Fig. 6b displays the progression of the normalized

absorbance curves for all three BSA concentrations. The solid lines show the corresponding sigmoidal Boltzmann equation curves from fitting to the data in Fig. 6a defined as: $y = A_2 + (A_1 - A_2) \cdot (1 + e^{(x-x0)dx^{-1}})^{-1}$ with the initial and final absorbance value $A_1$ and $A_2$, respectively, the transition temperature $x0$ as defined by the $x0$-value that corresponds to $y0 = (A_1 + A_2) \cdot 2^{-1}$ and the slope $dx$. They follow the same s-shape as seen from other experiments in literature with BSA[31] as well as other proteins[13], including decreasing transition temperatures $x0$ with increasing BSA-concentration (see also Schwaighofer et al. for e.g., the protein poly-l-lysine[13]).

Such a behavior was shown to be a function of the heating rate during the denaturation process. We therefore analyzed the corresponding heating for all three concentrations, confirming almost identical heating rates as shown in Supplementary Fig. 4.

## Sensor operation under real-life protein conditions

The $D_2O$ matrix provides very similar conditions for observing the temperature-induced BSA denaturation process as compared to fully

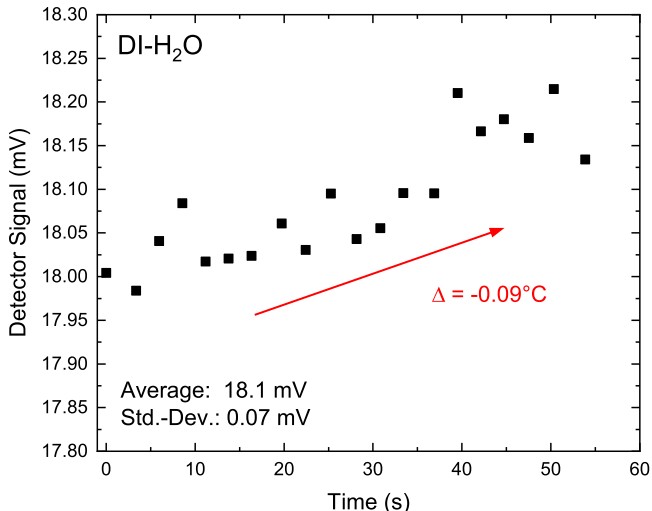

**Fig. 7 | Submersion experiment in native biophysical conditions, i.e., water.** Detector Signal when operating the QCLD sensor in DI-H$_2$O for about 1 min and increasing the temperature by about 0.1 ℃.

native biophysical conditions in H$_2$O. We therefore carefully selected this experiment, since the differences with H$_2$O are only a slight increase in the transition temperature for BSA denaturation[32]. Nevertheless, we want to demonstrate the full capability of our monolithic device to operate in a real-life protein matrix and therefore we conducted an additional submersion experiment in pure H$_2$O. As shown in Fig. 7, we submerged the whole sensor into water under operational bias with similar driving conditions as in D$_2$O, for approximately one minute, and monitored the detector signal. As H$_2$O absorbs the entire intensity of the optical mode, the remaining detector signal results from on-chip crosstalk, due to the compact nature of the device and could be used for crosstalk correction. An indication for proper sensor operation in aqueous solution can be seen from the slight temperature change of ~0.09 ℃ in the course of the 1-min submersion, and the simultaneous small increase of the detector signal (~0.2 mV), following a similar behavior as in deuterated solution. We want to stress, that this experiment was performed between two denaturation measurements, and we did observe no impact or negative effect on the sensor operation when comparing pre- and post-water submersion performance.

In the next step, realizing an on-chip QCLD-sensor for similar reaction monitoring experiments, as previously performed in D$_2$O, but operating in a highly-absorbing matrix like water, requires a re-designed device geometry. Much shorter plasmonic interaction lengths, typically in the order of 10–15 μm are needed in such a case and are part of our future work. The presented fundamental demonstration of monolithic device operation in water opens the door for using similar QCLD sensors, unlocking the whole field of on-chip mid-IR reaction monitoring of biochemical and pharmaceutical samples.

## Discussion

In conclusion, we show a next-generation of optical, fingertip-sized mid-IR lab-on-a-chip, suitable for sensitive and selective in situ real-time analysis of chemical reactions in liquids.

We analyzed the denaturation process of the protein BSA in a D$_2$O matrix with our sensor, which operates at the wavelengths of 1597 cm$^{-1}$ and 1620 cm$^{-1}$. The QCLD is based on monolithic integration of a QCL, a DLSPP waveguide, and a QCD on a single miniaturized chip. It allows in situ and online measurements in real time, probing in the latter case only microliter amounts of liquid. Its high-performance is demonstrated by a very low LOD of 75 ppm by weight (=0.0075% m v$^{-1}$) and it follows the Beer-Lambert law up to 9.23% m v$^{-1}$ of BSA concentration.

This is confirmed by a calibration line measurement throughout the whole concentration range spanning more than three orders of magnitude. In a protein denaturation measurement we reveal the typical sigmoidal s-shaped increase of absorbance with increasing BSA treatment temperature, together with a concentration-dependent transition temperature. The latter could be shown by performing dynamic denaturation measurements at concentrations of 20, 40, and 60 mg ml$^{-1}$.

In addition, the behavior of the QCLD-sensor, when submerged into a liquid like BSA in D$_2$O, was modeled through FEM-based simulations (COMSOL) of the plasmonic interaction section including the analyte. They show the excellent suitability of this type of monolithic sensor and materials for sensing in a D$_2$O matrix and for observing thermal denaturation processes in a wide range of BSA concentrations. This presents a suitable demonstration of an on-chip dynamical reaction monitoring in real time.

After the detailed analysis of our sensor in this work using a D$_2$O matrix, including a first demonstration of device operation in real-life protein conditions using a water matrix, the next step will be a full study of dynamical processes under biophysical conditions in H$_2$O. This will need the discussed re-designed and optimized plasmonic waveguide geometry, together with a careful selection of measurement wavelength(s) for avoiding the highest absorption peaks in water. With the results of the current work, including the outcome of the performed simulations, such an optimized design can now be straightforward implemented.

## Methods
### ATR measurements
FTIR absorption measurements of BSA were performed using a Bruker Tensor 37 FTIR spectrometer (Ettlingen, Germany) equipped with a Bruker Optics Platinum ATR module (diamond crystal, 1 mm$^2$ with single reflection) and a DLaTGS (deuterated lanthanum a-alanine doped triglycine sulfate) detector ($D^* = 6 \times 10^8$ cm $\sqrt{Hz}$ W$^{-1}$ at 9.2 μm). During the measurements, the spectrometer was constantly flushed with dry air for at least 10 min prior to data acquisition until water vapor absorption was sufficiently constant. Spectra were acquired with a resolution of 4 cm$^{-1}$ in double-sided acquisition mode; the mirror velocity was set to 20 kHz. A total of 26 scans (measurement time: 60 s) were averaged per spectrum, which was calculated using a Blackman-Harris 3-term apodization function and a zero-filling factor of 2. All spectra were acquired at 25 ℃. The recorded ATR-FTIR spectra were treated with advanced ATR correction and analyzed using the software package OPUS 8.1 (Bruker, Ettlingen, Germany). For quantitative measurements, 30 μl of BSA solution in D$_2$O at concentrations between 1 and 50 mg ml$^{-1}$ were placed on the ATR crystal and FTIR spectra were recorded. The sensitivity, or slope m of linear regression, was used for the calculation of the limit of detection (LOD), as follows: LOD $= 3 \cdot$ RMS-noise $\cdot$ m$^{-1}$. The root mean square (RMS) noise of the instrument was measured in the spectral region between 1550 and 1650 cm$^{-1}$ (with the respective number of scans) and the slope of the calibration line was determined at 1597 cm$^{-1}$.

### General procedure of on-chip QCLD measurements
All measurements (concentration- and denaturation-series) with the monolithic QCLD-sensor follow the same routine: first a reference measurement in pure D$_2$O is performed (averaging time per datapoint: 2 s, typical data acquisition time: 30–60 s) as a baseline, directly followed by the BSA-measurement. This results in an individual and accurate reference measurement for every BSA-containing measurement. In the case of the denaturation measurement in the microfluidic cell, we flushed the cell for at least 60 s with the analyte at the fixed concentration. After every measurement we purged, i.e., cleaned, the sensor for multiple minutes with D$_2$O to remove residuals of the previous BSA exposure from the chip-surface.

It is worth noting, that the chip was submerged and operated in liquid solution for more than 40 h in total. This includes calibration measurements and purging with pure $D_2O$ as well as measurements with BSA in $D_2O$. The total operation time in a liquid separates in times with and without applied bias.

### On-chip QCLD denaturation measurements

On-chip absorption measurements of BSA were performed based on the following 3-step routine:

(i) Heating: a beaker containing 35 ml of BSA dissolved in $D_2O$ is prepared with three different concentrations: 20, 40, and 60 mg ml$^{-1}$. The analyte is then constantly heated (heating rate: ~0.1 °C) from ~20 °C to ~90 °C while being continuously pumped by a peristaltic pump (Ismatec Reglo ICC, 3 channels, 8 rolls) at a rate of 1 ml min$^{-1}$ to the cooling part of the setup using suitable microfluidic tubings.

(ii) Cooling: for fast cooling of the liquid analyte to the measurement temperature of 21 °C, the microfluidic tube is guided through a beaker with deionized-$H_2O$ thermally stabilized to ~ 20 °C. Due to the small volume of the BSA-solution in the microfluidic tube, a few seconds in the cooling liquid are enough to efficiently cool it down, even from the maximum heating temperature of 90 °C. This is confirmed by no observable temperature rise within the microliter-scale measurement cell for any of the applied temperatures of the heating bath.

(iii) Driving & Measuring: finally, the liquid is pumped into the custom-made microfluidic Aluminum cell (volume: 60 µl) which is mounted on top of the sensor chip to demonstrate its microliter-measurement capabilities. While temperature-stabilizing the whole cell to 21 °C, the measurement is performed by biasing the corresponding QCL that operates at 1620 cm$^{-1}$ (pulses: 100 ns, repetition rate: 5 kHz, Avtech AVL-2-B pulse generator) and reading the on-chip QCD signal using a 350 MHz oscilloscope (Teledyne LeCroy HDO4034 2.5 GSPS). The sample is then pumped out of the microliter cell and is disposed of.

### Reporting summary

Further information on research design is available in the Nature Research Reporting Summary linked to this article.

## Data availability

The data generated in this study have been deposited in the Zenodo database under accession code https://doi.org/10.5281/zenodo.6930083.

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

## Acknowledgements

Fruitful discussions with W. Schrenk and E. Gornik are greatly acknowledged. We thank A. Linzer and I.C. Doganlar for expert technical assistance. B.H. and M.D. received funding from the EU Horizon 2020 Framework Program (project cFlow, No. 828893). This work was funded from the COMET Center CHASE (project No 868615) within the COMET - Competence Centers for Excellent Technologies program by the BMK, the BMDW and the Federal Provinces of Upper Austria and Vienna. The COMET program is managed by the Austrian Research Promotion Agency (FFG). B.H. acknowledges funding by the Austrian Science Fund FWF (M2485-N34) and A.S. through (project no. P32644-N). P.L.S. acknowledges the support received from CAPES-Brazil (88887.477460/2020-00). CzechNanoLab project LM2018110 funded by MEYS CR is gratefully acknowledged for the financial support of the measurements at CEITEC Nano Research Infrastructure. A.M.A. acknowledges funding from EOARD/AFOSR (FA8655-22-1-7170) and from the FFG under grant agreement number (883941, "Green Sensing MIR").

## Author contributions

B.H., F.P., L.L., P.L.S., and A.S. designed the experiments and performed the liquid measurements; P.L.S. characterized the quantum cascade devices; H.D. and A.M.A. grew the quantum cascade structures; D.R. and B.S. fabricated the devices; M.D. performed numerical simulations; B.H. and A.S. analyzed the results; B.H. wrote the manuscript with editorial input from F.P., A.S., A.M.A., B.L., and G.S.; all authors read the manuscript and commented on the paper.

## Competing interests

The authors declare no competing interests.
