## [Peer Review File · Nature Communications]

A mid-infrared lab-on-a-chip for dynamic reaction monitoringREVIEWER COMMENTS

Reviewer #1 (Remarks to the Author):

Review of the manuscript: A mid-infrared lab-on-a-chip for dynamic reaction monitoring by Hinkov et al.

Hinkov and coauthors present an interesting manuscript on the development of a sub-miniature MID-IR spectrometer based on a quantum cascade laser (QCL), a quantum cascade detector (QCD), and a plasmonic waveguide. This indeed demonstrates the great potential of quantum cascade laser and detector technologies and will drive the future development of QC-technologies in sensing applications and biomedical devices. They validate their approach with the analysis of a standard test protein, bovine serum albumin (BSA), by comparing concentration measurements with a reference analysis from FT-IR instrumentation combined with attenuated reflection technology (ATR). Overall, this is impressive and should thus be published.

In detail, the technological description is well documented, together with simulation results that support the concept.

The only deficit of this manuscript is the solvent chosen for validation with BSA: The authors selected heavy water (D₂O) to dissolve BSA, which allows them to perform the measurement at a higher effective path length. However, proteins in D₂O are far from biophysical reality and have little to do with real proteins in a relevant biophysical matrix. This is indeed an old-fashioned way-out in infrared spectroscopy whenever researchers are afraid of the high H-O-H background of standard aqueous buffers. It would have been much more convincing if the authors had designed the plasmonic waveguide for an effective pathlength of 10-15 μm maximum. If so, a measurement in a normal aqueous matrix also containing buffering substances (with some additional IR absorbance) would have been possible, and could have demonstrated the real practical application value.

What to do? I am far from asking the authors to restart their waveguide design and fabrication to meet the requirements for a measurement in H₂O buffers. I am convinced that this is feasible and will be one of the next steps the authors perform. However, they should add a paragraph on this issue (sort of „...we apologize for not using real life protein conditions“). The Reader will forgive them: Their interest is first in highly attractive semiconductor physics and (at least for the minority of the authors) second in infrared bioanalytics.

With such a paragraph added, the manuscript should be published, and, I bet, will find great attention among the Readers of Nature Communications.

Reviewer #2 (Remarks to the Author):

This is an interesting manuscript evaluating the performance of an IR-lab-on-chip system for protein analysis. However, as the sensing system itself has already been reported by some of the authors before (see several publications by B. Schwarz et al., e.g., Nat. Comm 2014), this manuscript should not be published in Nat. Comm. due to lack of novelty. Also, the analysis of BSA in D₂O is not of practical relevance, albeit a nice model example. However, for demonstrating real-world utility, measurements in H₂O are mandatory. In summary, this is a nice applied study, however, lacks in novelty and practical relevance. Therefore, it is much more suitable for publication in a more specific journal such as Applied Spectroscopy, as even for a more analytical journal it requires data in a relevant - i.e., aqueous - protein matrix.

Reviewer #3 (Remarks to the Author):

This is a solid and very well executed piece of device development work. Care has been taken to ensure that the understanding of the way the device is working is correct, and there is an impressive amount of characterisation and modelling work to underpin the conclusions. It is convincing and significant.

Chris Phillips.

Reviewer #1 (Remarks to the Author):

Answer from the Authors, citations from the new version of the manuscript

Review of the manuscript: A mid-infrared lab-on-a-chip for dynamic reaction monitoring by Hinkov et al.

Hinkov and coauthors present an interesting manuscript on the development of a sub-miniature MID-IR spectrometer based on a quantum cascade laser (QCL), a quantum cascade detector (QCD), and a plasmonic waveguide. This indeed demonstrates the great potential of quantum cascade laser and detector technologies and will drive the future development of QC-technologies in sensing applications and biomedical devices. They validate their approach with the analysis of a standard test protein, bovine serum albumin (BSA), by comparing concentration measurements with a reference analysis from FT-IR instrumentation combined with attenuated reflection technology (ATR). Overall, this is impressive and should thus be published.

In detail, the technological description is well documented, together with simulation results that support the concept.

The only deficit of this manuscript is the solvent chosen for validation with BSA: The authors selected heavy water (D₂O) to dissolve BSA, which allows them to perform the measurement at a higher effective path length. However, proteins in D₂O are far from biophysical reality and have little to do with real proteins in a relevant biophysical matrix. This is indeed an old-fashioned way-out in infrared spectroscopy whenever researchers are afraid of the high H-O-H background of standard aqueous buffers. It would have been much more convincing if the authors had designed the plasmonic waveguide for an effective pathlength of 10-15 μm maximum. If so, a measurement in a normal aqueous matrix also containing buffering substances (with some additional IR absorbance) would have been possible, and could have demonstrated the real practical application value. What to do? I am far from asking the authors to restart their waveguide design and fabrication to meet the requirements for a measurement in H₂O buffers. I am convinced that this is feasible and will be one of the next steps the authors perform. However, they should add a paragraph on this issue (sort of „...we apologize for not using real life protein conditions &“). The Reader will forgive them: Their interest is first in highly attractive semiconductor physics and (at least for the minority of the authors) second in infrared bioanalytics.

With such a paragraph added, the manuscript should be published, and, I bet, will find great attention among the Readers of Nature Communications.

Dear Reviewer #1,

Thank you very much for your thorough and detailed analysis of our work. Your review clearly helps to further improve the quality of our manuscript and we are thankful for your overall positive assessment.

We agree with your assessment, that the focus of this work is on the first full analysis of our monolithic QCLD sensor in a dynamic reaction monitoring experiment. We designed our on-chip sensor to operate in heavy water, because we strongly benefit from its much lower background absorption. It enables accurate sensor characterization and demonstration of its full potential towards important sensor figures-of-merit including long interaction lengths, low analyte concentrations, accurate determination of the sensor LOD and coverage of more than three orders of BSA concentrations. We also tested our QCLD sensor in H₂O, but as expected the 48 μm interaction length along the plasmonic waveguide is too long and all the optical mode is absorbed. Nevertheless, this additional experiment was still useful in two ways: first of all, it demonstrates the fundamental capability to operate our device under fully native protein conditions in H₂O. And second, it simultaneously allows us to extract the electrical crosstalk signal resulting from our compact on-chip geometry. The latter is

possible exactly because the entire optical signal gets absorbed. We now also refer to these points in the updated manuscript, when discussing the use of heavy water instead of normal water.

Generally, we agree with your observation that it would have been of practical relevance to demonstrate protein folding experiments even in normal water. As you say, and we fully agree with you, this is possible by reducing the pathlength of the plasmonic waveguide to about 10-15 μm , which is exactly what we will do as part of a next study. Such a complete adaption to H_2O needs a redesign of the plasmonic waveguide and fabrication of new devices, together with an new set of experiments.

However, this fact does not diminish the novelty and relevance of our work, which we see primarily in the successful demonstration of a chip-based QCLD sensor that can be immersed into water (heavy and normal) for performing label-free protein folding studies *in-situ* and in real time, while only probing microliters of sample liquid.

We would also like to stress, that temperature-induced folding of BSA only differs slightly in heavy and normal water, as reported by e.g. Zhou et al., *Vib. Spectrosc.* 92, 273 (2017). Comparison of our data with previous studies shows good agreement and thus strengthens our claim of being able to observe protein folding *in-situ* using the fully immersed QCLD sensor.

Having read your comments we believe that we share a very similar view on how to move forward. We agree that we should comment in more detail in the manuscript the use of heavy water and also discuss the adaption needed for realizing similar experiments in normal water. We have therefore updated the manuscript accordingly (see below).

From the introduction of the updated manuscript (new parts in red):

“In this work, we present a fully monolithic integrated mid-IR sensor, that combines all of the above features into a single, miniaturized device. Through the combination of the laser, interaction region, and detector on one chip, and avoiding typical diffraction limitations of conventional chip-scale photonic systems [6,25] by exploiting plasmonic waveguides [26–28], we realize a finger-tip sized (<5x5 mm²) next-generation rapid liquid sensor. Simulation results confirm the preservation of plasmonic capabilities in a liquid environment and enable the use of spectrally optimized QCLDs, i.e. devices that emit and detect similar-wavelength photons [29].

We aim to accurately characterize our QCLD sensor and extract its relevant figures-of-merit, together with demonstrating its capability to monitor changes in the secondary structure of proteins in real time. For these experiments, we employ bovine serum albumin (BSA) as a water-soluble monomeric protein (see Fig. 1(a) and Supplementary A). BSA is frequently used in fundamental biophysical studies, such as the investigation of thermal denaturation [30–32] that leads to changes in mid-IR absorption in the amide I region of proteins. Investigations of thermal denaturation of BSA were previously regularly performed in both, H_2O and D_2O [30–34], providing a rich data set, especially for those experiments in heavy water.

*Heavy water is often employed for protein analysis to avoid overlap of the HOH bending band of water with the protein amide I region in mid-IR spectroscopy. This creates an open spectral window in this range with reduced absorption, enabling larger interaction lengths [12, 31–41]. Generally, heavy water can provide some deviations from fully native biophysical conditions. It was found that exposure of proteins to D_2O may influence the length and strength of hydrogen bonds and may cause changes in protein dynamics [42] as well as protein denaturation [43]. But such different properties of D_2O can also be exploited in the analysis of specific biological samples, e.g. for significantly slowing down protein-solvent interactions in living cells for *in-vitro* experiments [44].*

For the here reported experiment in the thermal denaturation of BSA, it was found that the structural transitions of BSA were similar in H_2O and D_2O , with the single difference of a lower transition

temperature in heavy water [34]. Working in D₂O thus allows us to record data with a high signal-to-noise ratio and to exploit the full potential of our sensor concept in dynamic reaction monitoring. It also enables to much better harness the plasmonic-sensing concept, which allows optical mode propagation on the tens-to-hundreds of micrometers length scales. Our on-chip sensor features a sample interaction length of ~48 μm, which facilitates the analysis of BSA in D₂O over a wide concentration range, covering more than three orders of magnitude, from ~75 μg ml⁻¹ to >92 mg ml⁻¹. In contrast, using highly absorbing H₂O buffer typically limits path lengths to a maximum of 10 μm in the case of low intensity FTIR based experiments [45] and to approximately 25 μm [46] when performing high intensity QCL-based transmission measurements, with the consequence of a significantly reduced limit of detection (LOD).

In particular, we performed two types of measurements in our study. We determined the sensor calibration line and performed thermal denaturation experiments, monitoring the related change of the protein secondary structure [12, 13, 31, 33, 34,47]. Since our work includes an analysis of the sensor performance using optical Finite Element (FEM) simulations with the commercial software COMSOL, we can also theoretically confirm its excellent suitability for in-situ operation in a liquid matrix. Finally, we determine experimentally important analytical figures-of-merit including: 1) LOD, 2) sensor-linearity with analyte concentration, 3) the accessible concentration-range and sample volume of our sensor, and 4) robustness against direct exposure to the analyte.

We complete our study by demonstrating the operation of our QCLD sensor when immersed into normal water (H₂O), as biophysically native and most relevant matrix. As a direct consequence of the fully absorbed intensity of the optical mode in water, due to the large effective penetration depth of the sensor, we could simultaneously use the remaining measured detector signal from this experiment for electrical crosstalk correction.”

New section in the updated manuscript (new parts in red):

“5. Sensor Operation under Real-Life Protein Conditions

The D₂O matrix provides very similar conditions for observing the temperature-induced BSA denaturation process as compared to fully native biophysical conditions in H₂O. We therefore carefully selected this experiment, since the differences with H₂O are only a slight increase in the transition temperature for BSA denaturation [34]. Nevertheless, we want to demonstrate the full capability of our monolithic device to operate in a real life protein matrix and therefore we conducted an additional submersion experiment in pure H₂O. As shown in Fig. 7, we submerged the whole sensor into water under operational bias with similar driving conditions as in D₂O, for approximately one minute, and monitored the detector signal. As H₂O absorbs the entire intensity of the optical mode, the remaining detector signal results from on-chip crosstalk, due to the compact nature of the device and could be used for crosstalk correction. An indication for proper sensor operation in aqueous solution can be seen from the slight temperature change of ~0.09 °C in the course of the 1-minute submersion, and the simultaneous small increase of the detector signal (~0.2 mV), following a similar behavior as in deuterated solution. We want to stress, that this experiment was performed between two denaturation measurements, and we did observe no impact or negative effect on the sensor operation when comparing pre- and post-water submersion performance.

In the next step, realizing an on-chip QCLD-sensor for similar reaction monitoring experiments, as previously performed in D₂O, but operating in a highly-absorbing matrix like water, requires a re-designed device geometry. Much shorter plasmonic interaction lengths, typically in the order of 10 - 15 μm are needed in such a case and are part of our future work. The presented fundamental demonstration of monolithic device operation in water opens the door for using similar QCLD sensors, unlocking the whole field of on-chip mid-IR reaction monitoring of biochemical and pharmaceutical samples.”

Figure 7: Submersion experiment in native biophysical conditions, i.e. water. Detector Signal when operating the QCLD sensor in DI-H₂O for about 1 minute and increasing the temperature by about 0.1 °C.

By following your very helpful suggestions and addressing your comments together with adding those two sections to the manuscript, we believe that our work can be accepted for publication in Nature Communications and find your forecasted great attention among its readers.

Best regards,

Borislav Hinkov (in the name of the authors)

Reviewer #2 (Remarks to the Author):

Answer from the Authors, citations from the new version of the manuscript

This is an interesting manuscript evaluating the performance of an IR-lab-on-chip system for protein analysis. However, as the sensing system itself has already been reported by some of the authors before (see several publications by B. Schwarz et al., e.g., Nat. Comm 2014), this manuscript should not be published in Nat. Comm. due to lack of novelty. Also, the analysis of BSA in D₂O is not of practical relevance, albeit a nice model example. However, for demonstrating real-world utility, measurements in H₂O are mandatory. In summary, this is a nice applied study, however, lacks in novelty and practical relevance. Therefore, it is much more suitable for publication in a more specific journal such as Applied Spectroscopy, as even for a more analytical journal it requires data in a relevant - i.e., aqueous - protein matrix.

Dear Reviewer #2,

We want to thank you for your feedback on our work. Your review helps to further improve the quality of our manuscript. We understand, that we need to better explain the novelty of our work and why we chose to measure BSA in D₂O.

1. Novelty:

The QCLD principle was published before (as cited: [26,27]), as a mere proof-of-principle. However, this does not diminish the novelty and relevance of our work, which we see primarily in the successful demonstration of a chip-based QCLD sensor that can be immersed into water (heavy and normal) for performing label-free dynamic reaction monitoring experiments. This is demonstrated by *in-situ* protein folding studies in real time, probing microliters of liquid only. In addition, we show the first simulation work that confirms the preservation of the mid-IR plasmonic capabilities in a liquid environment, which enabled the realization of spectrally optimized QCLD devices with suitable plasmonic geometry (48 μm plasmonic interaction length) for measurements in D₂O.

They are operated in two typical analytical chemistry settings:

- i) *in-situ* monitoring by submerging the whole chip into the liquid sample
- ii) in-line measurements using a novel custom-made flow cell hosting 60 μl

Here, we also agree with the other two reviewers on their assessment of novelty and impact of our work.

Nevertheless, we understand the need to better highlight this in the manuscript and have therefore updated the manuscript accordingly:

From the introduction of the updated manuscript (new parts in red):

"In this work, we present a fully monolithic integrated mid-IR sensor, that combines all of the above features into a single, miniaturized device. Through the combination of the laser, interaction region, and detector on one chip, and avoiding typical diffraction limitations of conventional chip-scale photonic systems [6,25] by exploiting plasmonic waveguides [26–28], we realize a finger-tip sized (<5x5 mm²) next-generation rapid liquid sensor. Simulation results confirm the preservation of plasmonic capabilities in a liquid environment and enable the use of spectrally optimized QCLDs, i.e. devices that emit and detect similar-wavelength photons [29].

We aim to accurately characterize our QCLD sensor and extract its relevant figures-of-merit, together with demonstrating its capability to monitor changes in the secondary structure of proteins in real time. For these experiments, we employ bovine serum albumin (BSA) as a water-soluble monomeric protein (see Fig. 1(a) and Supplementary A). BSA is frequently used in fundamental biophysical studies, such as

the investigation of thermal denaturation [30–32] that leads to changes in mid-IR absorption in the amide I region of proteins. Investigations of thermal denaturation of BSA were previously regularly performed in both, H₂O and D₂O [30–34], providing a rich data set, especially for those experiments in heavy water.”

2. BSA in D₂O:

In mid-IR spectroscopy there is a long-standing tradition that fundamental studies on protein folding and bio-ligand interactions can be carried out in heavy water, being a valid surrogate for aqueous (H₂O) solutions. It has been repeatedly shown that changes in the secondary structure of proteins can be observed equally well in normal and heavy water. Due to the H-D exchange in the proteins' amide moiety, there are differences in the observed spectral changes, reaction kinetics and transition temperatures. The amide I band, which is mainly related to the C=O stretch vibration, is redshifted by only a few wavenumbers, while the amide II band, mainly related to the NH, respectively the N-D bending vibration, is redshifted by about one hundred wavenumbers. For protein folding, it is well known that spectral changes occurring in the amide I are of the highest diagnostic value. It was reported by Zhou et al. (Vib. Spectrosc. 92, 273 (2017)) that very similar reaction dynamics at slightly lower transition temperatures were observed in D₂O for the temperature-induced BSA denaturation. Thus, we can expect similar characteristics for our QCLD sensor measuring in either D₂O or in H₂O, including similar s-shaped and concentration-dependent denaturation curves, linear calibration lines (Beer-Lambert-Law), and analogous plasmonic behavior in the submerged sensor geometry.

The relevance of heavy water as routinely used solvent matrix in mid-IR based protein studies is further emphasized by numerous publications in recent years, see for example: *Dabrowska et al., Opt. Express 28(24), 36632 (2020)*; *Güler et al., Spectrochim. Acta A, 161, 8 (2016)*; *Lu et al., Analyst 140(3), 765 (2015)*; *Yang et al., Nat. Protoc. 10(3), 382 (2015)*; *De Meutter et al., Eur. Biophys. J. 50, 613 (2021)*; *Strazdaite et al., Langmuir 36(17), 4766 (2020)*.

The obtained results demonstrate that our developed on-chip QCLD sensor can *in-situ* monitor dynamic changes when submerged in water. It is significant that the observed reaction dynamics agree with previous studies by e.g. Zhou et al.

Nevertheless, we agree that we should comment in more detail on the use of heavy water in the manuscript and discuss the steps needed in terms of reduction of the plasmonic waveguide length to about 10-15 μm for realizing similar experiments in H₂O. We updated the manuscript accordingly.

Generally, we intend to demonstrate the first full analysis of our monolithic QCLD sensor in an *in-situ* dynamic reaction monitoring experiment. We now also better explain the substantial benefits of using heavy water over normal water with its much lower background absorption in this wavelength range. It enables accurate sensor characterization and demonstrates its full potential with important sensor figures-of-merit, including long interaction lengths, low analyte concentrations, accurate determination of the sensor limit of detection (LOD), and coverage of more than three orders of magnitude in BSA concentrations. We refer to these points in more detail in the updated manuscript.

We also tested our device in H₂O. As expected, the as designed, and for measurements in D₂O optimized, 48 μm long plasmonic interaction section is too long for protein folding experiments in H₂O, resulting in absorption of the entire optical signal. Nevertheless, this additional new experiment completes our study by demonstrating the operation of our QCLD sensor under native protein conditions in H₂O. In addition, this measurement allows us to extract the electrical crosstalk resulting from our compact on-chip sensor geometry.

These critical findings demonstrate the fundamental capability of our sensor for future experiments under fully native biophysical conditions in H₂O and are part of the updated manuscript (see below).

From the introduction of the updated manuscript (new parts in red):

“Heavy water is often employed for protein analysis to avoid overlap of the HOH bending band of water with the protein amide I region in mid-IR spectroscopy. This creates an open spectral window in this range with reduced absorption, enabling larger interaction lengths [12, 31–41]. Generally, heavy water can provide some deviations from fully native biophysical conditions. It was found that exposure of proteins to D₂O may influence the length and strength of hydrogen bonds and may cause changes in protein dynamics [42] as well as protein denaturation [43]. But such different properties of D₂O can also be exploited in the analysis of specific biological samples, e.g. for significantly slowing down protein-solvent interactions in living cells for in-vitro experiments [44].

For the here reported experiment in the thermal denaturation of BSA, it was found that the structural transitions of BSA were similar in H₂O and D₂O, with the single difference of a lower transition temperature in heavy water [34]. Working in D₂O thus allows us to record data with a high signal-to-noise ratio and to exploit the full potential of our sensor concept in dynamic reaction monitoring. It also enables to much better harness the plasmonic-sensing concept, which allows optical mode propagation on the tens-to-hundreds of micrometers length scales. Our on-chip sensor features a sample interaction length of ~48 μm, which facilitates the analysis of BSA in D₂O over a wide concentration range, covering more than three orders of magnitude, from ~75 μg ml⁻¹ to >92 mg ml⁻¹. In contrast, using highly absorbing H₂O buffer typically limits path lengths to a maximum of 10 μm in the case of low intensity FTIR based experiments [45] and to approximately 25 μm [46] when performing high intensity QCL-based transmission measurements, with the consequence of a significantly reduced limit of detection (LOD).

In particular, we performed two types of measurements in our study. We determined the sensor calibration line and performed thermal denaturation experiments, monitoring the related change of the protein secondary structure [12, 13, 31, 33, 34,47]. Since our work includes an analysis of the sensor performance using optical Finite Element (FEM) simulations with the commercial software COMSOL, we can also theoretically confirm its excellent suitability for in-situ operation in a liquid matrix. Finally, we determine experimentally important analytical figures-of-merit including: 1) LOD, 2) sensor-linearity with analyte concentration, 3) the accessible concentration-range and sample volume of our sensor, and 4) robustness against direct exposure to the analyte.

We complete our study by demonstrating the operation of our QCLD sensor when immersed into normal water (H₂O), as biophysically native and most relevant matrix. As a direct consequence of the fully absorbed intensity of the optical mode in water, due to the large effective penetration depth of the sensor, we could simultaneously use the remaining measured detector signal from this experiment for electrical crosstalk correction.”

the updated manuscript (new parts in red):

“5. Sensor Operation under Real-Life Protein Conditions

The D₂O matrix provides very similar conditions for observing the temperature-induced BSA denaturation process as compared to fully native biophysical conditions in H₂O. We therefore carefully selected this experiment, since the differences with H₂O are only a slight increase in the transition temperature for BSA denaturation [34]. Nevertheless, we want to demonstrate the full capability of our monolithic device to operate in a real life protein matrix and therefore we conducted an additional submersion experiment in pure H₂O. As shown in Fig. 7, we submerged the whole sensor into water under operational bias with similar driving conditions as in D₂O, for approximately one minute, and monitored the detector signal. As H₂O absorbs the entire intensity of the optical mode, the remaining detector signal results from on-chip crosstalk, due to the compact nature of the device and could be used for crosstalk correction. An indication for proper sensor operation in aqueous solution can be seen from the slight temperature change of ~0.09 °C in the course of the 1-minute submersion, and the

simultaneous small increase of the detector signal (~ 0.2 mV), following a similar behavior as in deuterated solution. We want to stress, that this experiment was performed between two denaturation measurements, and we did observe no impact or negative effect on the sensor operation when comparing pre- and post-water submersion performance.

In the next step, realizing an on-chip QCLD-sensor for similar reaction monitoring experiments, as previously performed in D_2O , but operating in a highly-absorbing matrix like water, requires a re-designed device geometry. Much shorter plasmonic interaction lengths, typically in the order of 10 - 15 μm are needed in such a case and are part of our future work. The presented fundamental demonstration of monolithic device operation in water opens the door for using similar QCLD sensors, unlocking the whole field of on-chip mid-IR reaction monitoring of biochemical and pharmaceutical samples.”

Figure 7: **Submersion experiment in native biophysical conditions, i.e. water.** Detector Signal when operating the QCLD sensor in DI-H₂O for about 1 minute and increasing the temperature by about 0.1 °C.

We believe, that this suitably addresses your comments and concerns and the updated manuscript demonstrates its relevance to the readers of Nature Communications.

Best regards,

Borislav Hinkov (in the name of the authors)

Reviewer #3 (Remarks to the Author):

Answer from the Authors, citations from the new version of the manuscript

This is a solid and very well executed piece of device development work. Care has been taken to ensure that the understanding of the way the device is working is correct, and there is an impressive amount of characterisation and modelling work to underpin the conclusions. It is convincing and significant.

Dear Reviewer #3,

We want to thank you very much for your thorough analysis of our work including the overall very positive assessment of our manuscript. Following your judgement, we believe that it can be accepted for publication in Nature Communications.

Best regards,

Borislav Hinkov (in the name of the authors)

REVIEWERS' COMMENTS

Reviewer #1 (Remarks to the Author):

In the revised version, the authors have addressed all relevant points raised by this reviewer. The manuscript can be published in its present form.